

# Global expansion of COVID-19 pandemic is driven by population size and airport connections

Marco Tulio Pacheco Coelho[1], João Fabrício Mota Rodrigues[1], Anderson Matos Medina[2], Paulo Scalco[3], Levi Carina Terribile[4], Bruno Vilela[2], José Alexandre Felizola Diniz-Filho[1] and Ricardo Dobrovolski[2]

[1] Departamento de Ecologia, Universidade Federal de Goiás, Goiânia, GO, Brazil
[2] Instituto de Biologia, Universidade Federal da Bahia, Salvador, BA, Brazil
[3] Faculdade de Administração, Economia, Ciências Contábeis (FACE), Universidade Federal de Goiás, Goiânia, GO, Brazil
[4] Laboratório de Macroecologia, Universidade Federal de Jataí, Jataí, GO, Brazil

## ABSTRACT

The pandemic state of COVID-19 caused by the SARS CoV-2 put the world in quarantine, led to hundreds of thousands of deaths and is causing an unprecedented economic crisis. However, COVID-19 is spreading in different rates at different countries. Here, we tested the effect of three classes of predictors, i.e., socioeconomic, climatic and transport, on the rate of daily increase of COVID-19 on its exponential phase. We found that population size and global connections, represented by countries' importance in the global air transportation network, are the main explanations for the early growth rate of COVID-19 in different countries. Climate and socioeconomics had no significant effect in this big picture analysis. Our results indicate that the current claims that the growth rate of COVID-19 may be lower in warmer and humid countries should be taken very carefully, risking to disturb well-established and effective policy of social isolation that may help to avoid higher mortality rates due to the collapse of national health systems.

## INTRODUCTION

With the worldwide spread of the novel Coronavirus Disease 2019 (COVID-19), caused by the SARS-CoV-2 virus, we are experiencing a declared pandemic. One of the largest preoccupations about this new virus is its notable ability to spread given the absence of any effective treatment, vaccine, and immunity in human populations. Epidemiologists quantify the ability of infectious agents to spread by estimating the basic reproduction number ($R0$) statistic (*Delamater et al., 2019*), which measures the average number of people each contagious person infects. The new coronavirus is transmitting at an average $R0$ between 2.7 and 3.2 (*Billah, Miah & Khan, 2020*; *Liu et al., 2020*), which is greater than seasonal influenza viruses that spread every year around the planet (median $R0$ of 1.28, *Biggerstaff et al., 2014*). To anticipate the timing and magnitude of public interventions

Corresponding author
Marco Tulio Pacheco Coelho,
marcotpcoelho@gmail.com

and mitigate the adverse consequences on public health and economy, understanding the factors associated with the survival and transmission of SARS-CoV-2 is urgent.

Because previous experimental (*Lowen et al., 2007*), epidemiological (*Shaman et al., 2010*; *Barreca & Shimshack, 2012*), and modeling (*Zuk, Rakowski & Radomski, 2009*) studies show the critical role of temperature and humidity on the survival and transmission of viruses, recent studies are testing the effect of environmental variables on SARS-CoV-2 (*Wang et al., 2020*; *Sajadi et al., 2020*; *Harbert, Cunningham & Tessler, 2020*; *Araujo & Naimi, 2020*) and forecasting monthly scenarios of the spread of the new virus based on climate suitability (*Araujo & Naimi, 2020*, *but see* *Carlson et al., 2020*). Although temperature and humidity affect the spread and survival of other coronaviruses (i.e., SARS-CoV and MERS-CoV, *Tan et al., 2005*; *Chan et al., 2011*; *Doremalen, Bushmaker & Munster, 2013*; *Gaunt et al., 2010*) using the current occurrences of SARS-CoV-2 cases to build correlative climatic suitability models without considering connectivity among different locations and socioeconomic conditions might be inadequate, especially considering that the definition of climatic niche of a respiratory virus, transmitted from person to person, is very challenging (*Carlson et al., 2020*).

Many factors might influence the distribution of diseases at different spatial scales. Climate might affect the spread of viruses given its known effect on biological processes that influences many biogeographical patterns, including the distribution of diseases and human behavior (e.g., *Murray et al., 2018*). Geographic distance represents the geographical space where the disease spread following the distribution of hosts and also explains biogeographic patterns (*Poulin, 2003*; *Nekola & White, 1999*; *Warren et al., 2014*). Socioeconomic characteristics of countries include population size, which represent a key epidemiological parameter that determines the rate and reach of pandemics (*Grassly & Fraser, 2008*) and other variables that represent a proxy for the ability to identify and treat infected people and for the governability necessary to make fast political decision and avoid the spread of new diseases (*Adler & Newman, 2002*; *Gilbert et al., 2020*; *Khalatbari-Soltani et al., 2020*). Finally, the global transportation network might surpass other factors as it can reduce the relative importance of geographic distance and facilitate the spread of viruses and their vectors (*Brockmann & Helbing, 2013*; *Pybus, Tatem & Lemey, 2015*). According to the *International Air Transport Association (2019)* more than 4 billion passengers made international travels in 2018. This amount of travelers reaching most of our planet's surface represents a human niche construction (*Boivin et al., 2016*) that facilitates the global spread of viruses and vectors (*Brockmann & Helbing, 2013*) in the same way it facilitated the spread of invasive species and domesticated animals over modern human history (*Boivin et al., 2016*).

The spread of SARS-CoV-2 from central China to other locations might be strongly associated with inter-country connections, which might largely surpass the effect of climate suitability. Thus, at this point of the pandemic, there is still a distributional disequilibrium that can generate very biased predictions based on climatic correlative modeling (*De Marco, Diniz-Filho & Bini, 2008*). Here we used an alternative macroecological approach (*Burnside et al., 2012*), based on the geographical patterns of early growth rates of the disease at country level, to investigate the drivers of the growth rates of COVID-19 in

its exponential phase. We studied the effect of environment, socioeconomic, and global transportation controlling for spatial autocorrelation that could bias model significance. By analyzing these factors, we show that the exponential growth rate of COVID-19 at global scale is explained mainly by population size and country's importance in the global transportation network.

## MATERIAL & METHODS

We collected the number of detected cases of COVID-19 per day from the John Hopkins (*Dong, Du & Gardner, 2020*) and European Centre for Disease Prevention and Control (*ECDC, 2020*). For each country we only used the "exponential" portion of the time series data (i.e., number of new people infected per day) and excluded days after stabilization or decrease in total number of cases (e.g., Fig. S1). Although we are aware that more complex logistic-like curves of growing cases are expected, simpler exponential growth rates are a simpler description of the expansion in early phases and in practice coefficients are indistinguishable from logistic when N << K. This procedure is also important to guarantee that only the early phase of the disease is analysed given that stabilization and decreasing in growth rates are caused both by natural population dynamics (following a logistic model) and by the interplay of different interventions made by each country, such as political and legal reinforcements of social distancing measures, including lockdown, obligatory mask use and others (*Chinazzi et al., 2020*; *Kraemer et al., 2020*; *Zhang et al., 2020*). Time series data are available for 204 countries, for which 65 had more than 100 cases recorded and for which time series had at least 30 days of exponential growth after the 100th case. We also performed the analysis considering countries with more than 50 cases, but it did not qualitatively change our results. Thus, we only show the results for countries with more than 100 cases.

We empirically modelled each time series using an exponential growth model for each country and calculated both the intrinsic growth rate (r) and the regression coefficient of the log growth series to be used as the response variable in our models. Because both were highly correlated (Pearson's $r = 0.97$, Fig. S2), we used only the regression coefficient to represent the growth rate of COVID-19 in our study.

To investigate potential correlates of the virus growth rate, we downloaded climatic and socioeconomic data of each country. We used climatic data represented by monthly average minimum and maximum temperature (°C) and total precipitation (mm) retrieved from the WorldClim database (https://www.worldclim.org) (*Fick & Hijmans, 2017*). We used monthly data for 2018, the most recent year available in WorldClim (*Fick & Hijmans, 2017*; *Harris et al., 2014*). We extracted climatic data from the months of January, February, March, and December to represent the climatic conditions of the winter season in the Northern Hemisphere and the summer season in the Southern Hemisphere. Temperature and precipitation are used here because of their critical role on virus transmissions (*Lowen et al., 2007*; *Shaman et al., 2010*; *Barreca & Shimshack, 2012*; *Zuk, Rakowski & Radomski, 2009*) and because of the recent investigations about its potential effect on the spread of COVID-19 (*Araujo & Naimi, 2020*; *Harbert, Cunningham & Tessler, 2020*). In addition, the

predefined time period, winter in the northern hemisphere and summer in the southern hemisphere, represents the seasons in which the virus started to spread in the different hemispheres. From these data, we computed the mean value of climatic variables across each country. Finally, minimum, and maximum temperatures were combined to estimate monthly mean temperature for December, January, February, and March, which was used in the model along with total precipitation for the same months. However, using different combinations of these variables (i.e., using means of minimum or maximum temperatures, as well as minimum or maximum for each month) did not qualitatively affect our results.

We extracted socioeconomic data for each country. Human Development Index (HDI) rank, mean number of school years in 2015, gross national income (GNI) per capita in 2018 population size in 2015 and average annual population growth rate between 2010–2015 were used in our study and downloaded from the United Nations database (http://hdr.undp.org/en/data) and from the World Inequality Database (https://wid.world). We also obtained a mean value of investments in health care by averaging the annual investments in health care in each country between 2005–2015 obtained from the World Health Organization database (http://apps.who.int/gho/data/node.home). Due to the strong collinearity among some of these predictors, HDI rank and mean number of school years were removed from our final model.

Finally, we also downloaded air transportation data from the *Openflights.org database (2014)* database regarding the airports of the world, which informs where each airport is located including country location (7,834 airports), and whether there is a direct flight connecting the airports (67,663 connections). We checked the Openfligths database to make the airports and connections compatible by including missing or fixing airport codes and removing six unidentified airport connections resulting in a total of 7,834 airports and 67,657 connections. We used this information to build an air transportation network that reflects the existence of a direct flight between the airports while considering the direction of the flight. Thus, the airport network is a unipartite, binary, and directed graph where airports are nodes and flights are links (Fig. 1, Fig. S3). In the following step, we collapsed the airports' network into a country-level network using the country information to merge all the airports located in a country in a single node (e.g., United States had 613 airports that were merged in a single vertex representing the country). The country-level network (Fig. 1, Fig. S3) is a directed weighted graph where the links are the number of connections between 226 countries which is collapsed for the 65 countries that had more than 100 cases and for which time series data had at least 30 days after the 100th case . Afterward, we measured the countries centrality in the network using the Eigenvector Centrality (*Bonacich, 1987*), that weights the importance of a country in the network considering the number of connections with other countries and how well connected these countries are to other countries—indirect connections. All networks analyses were generated using package *igraph* (*Csardi & Nepusz, 2006*).

We evaluated the relationship between the predictors (climatic, socioeconomic and transport data) and our growth rate parameter (the regression coefficient of the log-transformed growth series) using a standard multiple regression (OLS) considering the distribution of the original predictors as well as the normality of residuals. Moreover,

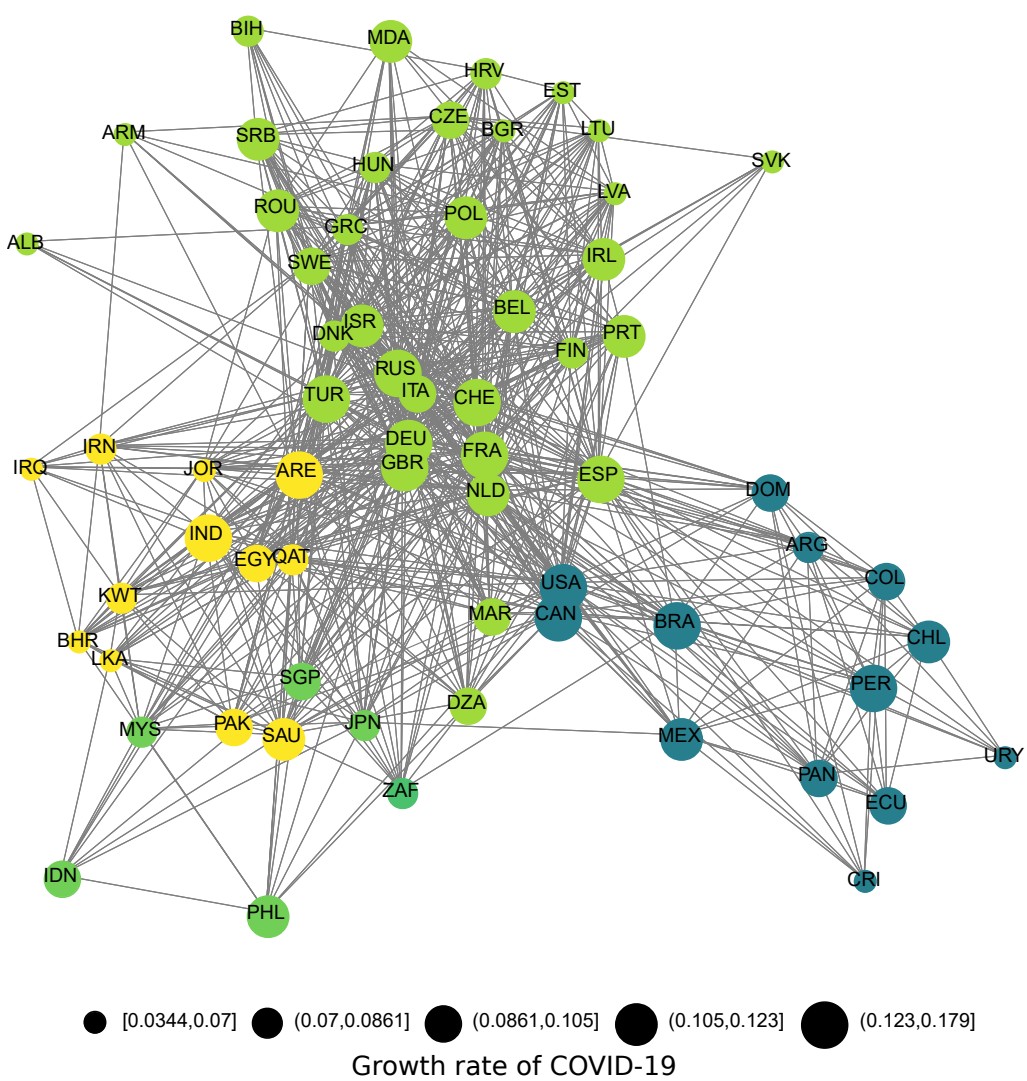

**Growth rate of COVID-19**

[0.0344,0.07] (0.07,0.0861] (0.0861,0.105] (0.105,0.123] (0.123,0.179]

**Figure 1** **Air transportation network among 65 countries that had more than 100 cases and for which time series data had at least 30 days after the 100th case.** Different colours represent modules of countries that are more connected to each other. Different sizes of each node represent the growth rate of COVID-19 estimated for each country.

OLS residuals were inspected to evaluate the existence of spatial autocorrelation that could upward bias the significance of predictor variables on the model using Moran's correlograms (*Legendre & Lengedre, 2013*). Prior to the analysis, we applied logarithmic (mean precipitation, total population size, and network centrality) and square root (mean health investments) transformations to the data to approximate a normal distribution.

## RESULTS

The models used to estimate COVID-19 growth rate on different countries showed an average $R^2$ of 0.91 (SD = 0.04), varying from 0.78 to 0.99, indicating an overall excellent

## Growth Rate

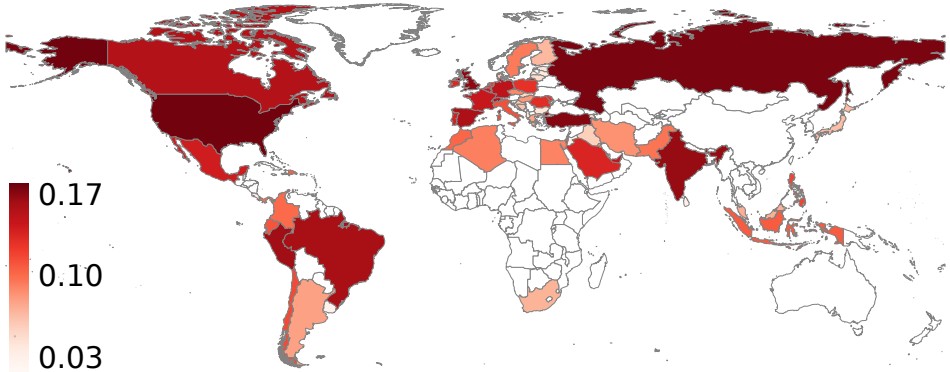

**Figure 2** **Geographical pattern of the early growth rate of COVID-19 in different countries.** Growth rate is represented by the regression coefficient of the log growth series.

**Table 1** **Model statistics for all variables used in the study.**

|  | Standardized estimate | Estimate | Std error | *t* value | *P*-value |
|---|---|---|---|---|---|
| *Intercept* |  | 0.074 | 0.023 | 3.232 | 0.002 |
| *Eigenvector Centrality* | 0.387 | 0.009 | 0.003 | 2.812 | **0.006** |
| *Gross National Income* | 0.264 | 0.000 | 0.000 | 1.764 | 0.083 |
| *Population Size* | 0.519 | 0.011 | 0.002 | 4.487 | **<0.001** |
| *Annual population growth* | 0.096 | 0.003 | 0.004 | 0.658 | 0.513 |
| *Heath investment* | −0.168 | 0.000 | 0.000 | −1.140 | 0.259 |
| *Mean Temperature* | −0.208 | −0.001 | 0.000 | −1.695 | 0.095 |
| *Mean Precipitation* | 0.184 | 0.006 | 0.003 | 1.730 | 0.089 |

performance on estimating growth rates. The geographical patterns in the growth rates of COVID-19 cases do not show a clear trend, at least in terms of latitudinal variation, that would suggest a climatic effect at macroecological scale (Fig. 2).

We build one model including only climate, which explained only 0.03% of the variation on growth rates. When we added socioeconomic variables, the $R^2$ increased to 53.95%. Finally, when we added country centrality (i.e., country importance in the global transportation network) as a predictor, the $R^2$ increased to 59.56%. In this model, only population size and country centrality had positive and significant effects (Fig. 3, Table 1). Thus, exponential growth rates increased *strongly* in response to countries population size and their importance in the transportation network (Table 1, Fig. 3). Statistical coefficients were not upward biased by spatial autocorrelation.

## DISCUSSION

At global scale, Gross National Income, annual population growth, investment in healthcare, mean temperature and mean precipitation had no significant effect on the
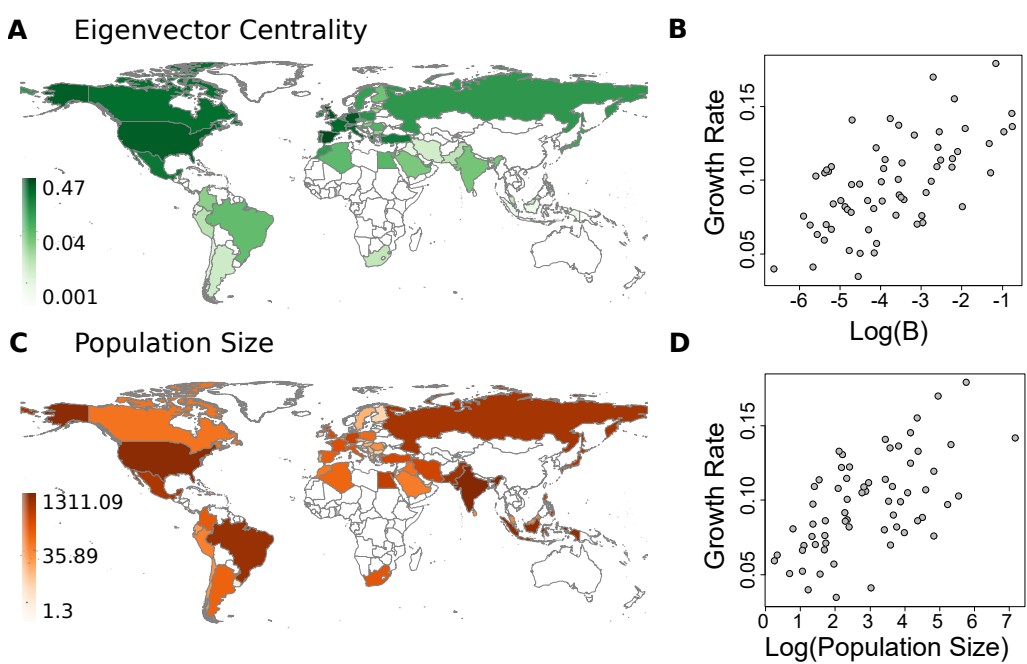

**Figure 3** **Spatial patterns of predictors and their relationship with COVID-19 growth rates.** Countries importance in the global transportation network (A) and population size (C) are strongly associated with early growth rates of COVID-19 across the world (B and D).

exponential phase of COVID-19. Population size and countries importance in the global transportation network have key roles on the growth rate of COVID-19.

Population size is a critical factor in epidemiological outbreaks and faster growths of COVID-19 were reported in cities with larger populations (*Stier, Berman & Bettencourt, 2020*). Here, we observe the same pattern at country level. Faster spread in regions with larger populations have been explained by the interaction of frequent trades and people exchanges, and the difficulty to control early outbreaks within larger populations (*Jaffe, Vera & Jaffe, 2020*; *Harbert, Cunningham & Tessler, 2020*; *Stier, Berman & Bettencourt, 2020*). Because of the multiple infection routes and faster spread in larger populations, recent discussions emphasize the need to implement more aggressive social distancing policies in regions with larger populations (*Stier, Berman & Bettencourt, 2020*). However, not only population size explains the exponential growth of COVID-19 in different countries but also how central a country is in the global transportation network.

Network centrality measures are widely used to discover distinguished nodes on networks, including epidemiological networks (e.g., *Madotto & Liu, 2016*). Our findings reinforce the importance of propagule pressure on disease dissemination (*Tian et al., 2017*; *Chinazzi et al., 2020*). Aerial transportation is an important predictor of COVID-19 dissemination in China (*Kraemer et al., 2020*), Brazil (*Ribeiro et al., 2020*), and Mexico (*Datillo et al., 2020*). Countries characterized by higher centrality in the global transportation network represent distinguished nodes, in terms of how well they are directly and indirectly connected to other countries. These countries are the ones that are

more prone to receive higher number of infected individuals in different regions of their territory, which can potentially contribute to the velocity of the initial spread of the disease.

The rapid international spread of the severe acute respiratory syndrome (SARS) from 2002 to 2003 led to extensively assessing entry screening measures at international borders of some countries (*Bell et al. 2004*; *John et al., 2005*). However, it is important to note that SARS-CoV-2 can spread from pre-symptomatic and asymptomatic individuals (*Gandhi, Yokoe & Havlir, 2020*; *Bai et al., 2020*). Thus, entry screening measures at international borders might be only partially effective to identify symptomatic individuals, but not effective to stop de disease at international borders. Even for diseases that could be stopped by identifying symptomatic travellers, there is no consensus of the effective and accurate tools to be used in airports across the globe (*Sun et al., 2017*). Finally, how effective airports closures were in different countries to decrease or stabilize the spread of COVID-19 still needs to be tested in different countries and is beyond the scope of this paper. However, after local transmissions are identified, we would expect that airport closures are less effective than any other measure taken by governments, such as increasing social distancing, tracking and isolating infected individuals (see *Chinazzi et al., 2020*).

When discussing and modelling the effect of climate on SARS CoV-2 it is important to remember that modern human society is a complex system composed by strongly connected societies that are all susceptible to rare events. It is also critical to consider the negative correlations between climate and local or regional socioeconomic conditions (i.e., inadequate sanitary conditions and poor nutritional conditions) that could easily counteract any potential climatic effect at local scales, such as lower survival rates of viruses exposed to high humidity, temperatures and high UV irradiation (*Wang et al., 2020*; *Duan et al., 2003*). Our analyses call attention to the case of Brazil, a well-connected and populated tropical country that presents one of the highest increase rates of COVID-19 in its exponential phase. If decision makers consider yet unsupported claims that growth rates of COVID-19 in its exponential phase might be lower in warmer and humid countries, we might observe terrible scenarios unrolling in well-connected and populated countries independent of their climatic conditions.

## CONCLUSIONS

Here, we show that countries' population size and importance in the global transportation network have key roles on the initial growth rate of COVID-19. We do not expect that our results using a macroecological approach at a global scale would have a definitive effect on decision-making in terms of public health in any particular country, province, or city.

However, we call the attention for the absence of effects of climatic variables on the exponential phase of COVID-19 that is surpassed by how distinguished a country is in the air transportation network and by their population size. Thus, claims that the growth of COVID-19 might be lower in warmer and humid countries based on climate suitability models should be taken very carefully, risking to disturb well-established and effective policy of social isolation that may help to avoid higher mortality rates due to the collapse in national health systems.

## ACKNOWLEDGEMENTS

We thank Thiago F. Rangel, André Menegotto, Robert D. Morris and Marcus Cianciaruso for their constructive comments on earlier versions of the manuscript. We also thank the editor Diogo Provete and the reviewer Blas M. Benito for valuable suggestions during the revision of this study.

### Funding

This paper was developed in the context of the human macroecology project on the National Institute of Science and Technology (INCT) in Ecology, Evolution and Biodiversity Conservation, supported by CNPq (grant 465610/2014-5) and FAPEG (grant 201810267000023). José Alexandre Felizola Diniz-Filho, Ricardo Dobrovolski, and Levi Carina Terribile are also supported by CNPq productivity scholarships. The funders had no role in study design, data collection and analysis, decision to publish, or preparation of the manuscript.

### Grant Disclosures

The following grant information was disclosed by the authors:
The National Institute of Science and Technology (INCT) in Ecology.
CNPq: 465610/2014-5.
FAPEG: 201810267000023.

### Competing Interests

The authors declare there are no competing interests.

### Author Contributions

- Marco Tulio Pacheco Coelho conceived and designed the experiments, analyzed the data, prepared figures and/or tables, authored or reviewed drafts of the paper, and approved the final draft.
- João Fabrício Mota Rodrigues conceived and designed the experiments, authored or reviewed drafts of the paper, and approved the final draft.
- Anderson Matos Medina analyzed the data, prepared figures and/or tables, authored or reviewed drafts of the paper, and approved the final draft.
- Paulo Scalco and Ricardo Dobrovolski conceived and designed the experiments, authored or reviewed drafts of the paper, and approved the final draft.
- Levi Carina Terribile and Bruno Vilela analyzed the data, authored or reviewed drafts of the paper, and approved the final draft.
- José Alexandre Felizola Diniz-Filho conceived and designed the experiments, analyzed the data, authored or reviewed drafts of the paper, and approved the final draft.

### Data Availability

The data and R code are available in the Supplemental Files.

## Supplemental Information

Supplemental information for this article can be found online at http://dx.doi.org/10.7717/peerj.9708#supplemental-information.

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
