# Peer review of "Global expansion of COVID-19 pandemic is driven by population size and airport connections"

_PeerJ, doi:10.7717/peerj.9708_

## Round 0.1 · original submission · Major Revisions

This is a timely paper that documents how airports can contribute to spread SARS-Cov2 and could help decision makers to design better strategies to deal with the pandemic. Both reviewers were very positive about your manuscript, but also pointed out a few aspects that could be improved. I have also read carefully the paper and I second them on every comment. Please, see my comments directly in the pdf attached.
Reviewer #1 pointed out a few sentences that could be improved, while Reviewer #2 was more critical and I highly recommend authors to pay closer attention to his/her suggestions on the potential mechanisms behind correlations found in the linear models, especially regarding the role of airline traffic connection in general and climatic variables for tropical countries.

·

Basic reporting

The article fulfils all the criteria.

Experimental design

The article fulfils all the criteria.

Validity of the findings

The article fulfils all the criteria.

Additional comments

Congratulations to the authors for such an elegant and timely paper.

I only have a few minor suggestions, these are listed below.

Line 32 (abstract) I am not sure to what extent the sentence "Geographic distance and climate were significant barriers in the past but were surpassed by the human engine that allowed us to colonize most of our planet land surface." adds meaning here, but I cannot put my finger on the why. Probably it could help not to mix geographic distance and climate together since the second part of the sentence only refers to the former. Maybe the authors could just remove it, and the abstract wouldn't lose any meaning.

Line 58: As one of the authors of Chipperfield et al. (2020), I think my coauthors would appreciate you changing the reference to "Carlson et al. (2020)", which is the final published version of Chipperfield et al. 2020. However, and considering the conflict of interest I have here, please, feel free to proceed as you wish regarding this suggestion.

Just in case, the complete reference would be:

Carlson, C.J., Chipperfield, J.D., Benito, B.M., Telford, R.J., O'Hara, R.B. 2020. Species distribution models are inappropriate for COVID-19. Nature Ecology and Evolution. URL: https://www.nature.com/articles/s41559-020-1212-8, DOI: 10.1038/s41559-020-1212-8

Line 87: maybe the authors want to say "...to investigate the drivers of the growth rate of SARS-CoV-2".

Line 94: I believe "detected cases" better reflects what the authors want to say with "people infected", since the amount of people infected is usually larger than the number of detected cases, and the used dataset is based on the results of PCR tests.

Line 189: Maybe the authors want to start the paragraph with "Network centrality measures are widely used...", unless they are specifically referring to the actual measure used in the analysis. If that is the case, please, name it again, the reader might not remember which one was used.

Line 205: I suggest to replace "countries that invested less in health" with "countries with a lower investment in healthcare". This is not an important suggestion though!

Line 217: there should be a citation at the end of "...of what has been suggested by climate suitability models."

Line 218: I am not a native English speaker, and I am having troubles to understand the meaning of "ephemeral" in this context, so maybe a rewording would help here.

Lines 231 to 236: Asymptomatic transmission of SARS-CoV-2 could go through airport screenings (unless mandatory quarantine is implemented), so maybe the authors could comment on how even board control measures might not be enough to limit the spread of a pandemic like the one we are undergoing at the moment. There are several papers talking about presymptomatic and asymptomatic transmission, for example:

Gandhi, M., Yokoe, D.S., Havlir, D.V., (2020) Asymptomatic transmission, the Achilles' heel of current strategies to control Covid-19. The New England Journal of Medicine. URL: https://www.nejm.org/doi/full/10.1056/NEJMe2009758, DOI: 10.1056/NEJMe2009758

Bai, Y., Yao, L., Wei, T., 2020. Presumed asymptomatic carrier tranmission of Covid-19. JAMA 323(14) 1406-1407. URL: DOI: https://jamanetwork.com/journals/jama/article-abstract/2762028, 10.1001/jama.2020.2565

Figure 2: I imagine it will save a lot of empty space to put panel A on top of the other panels.

Blas M. Benito.

Reviewer 2 ·

Basic reporting

Other remarks

Lines 47-48: The range of R0 reported in the literature is more [2-4], numerous models found values greater than 3.

Lines 100-102: It would be interesting to show some example of the exponential portion in comparison with the 30 days of the data.

Lines 105: For an analysis of an epidemic, it would be better to use the intrinsic growth rate (r) than the regression coefficient of the log growth series

Lines 136-143: About the airport network and the collapse of the airports’ network into a country-level network. This means that USA, China or Brazil is just one node in the network as Fiji, Suriname or Guyana. Then what are the consequences of the different population size between the countries of the network? Wouldn't it be better to standardize the connectivity of each country according to its population or some other economic variable?

Lines 136-143: What has been the role of airport closures and the sharp reduction in air traffic? How is this taken into account?

Lines 136-143: If you are flying from Europe to Australia, you stop in Thailand or in Singapore. How is this processed in the air traffic data? If I remember correctly just the IATA data take into account the different air stops for a given route.

Lines 154-156: What is the sensitivity of the results to the transformations used?

Results: Nothing is said about the link between the exponential growth rate of the epidemic at the beginning of the epidemic and the epidemic size. This would shed some light on the analysis

Line 159: “The models used to estimate COVID-19 growth rate on different countries” The growth rate or the regression coefficient of the log growth series as indicated in the Methods? And this question must be answer at each time of the expression “growth rate” is used in the rest of the manuscript.

Lines 185-188: “… we show that countries’ importance in the global transportation network has a key role on the severity of COVID-19 pandemic in different countries as it is strongly associated with the growth rates of the disease…” It’s not clear what is the meaning of the word “severity” in this sentence. Is it linked to mortality? But in this study, just the initial growth rate of the epidemic has been analyzed.

Lines 196: “secondary transmissions”: Definition?

Line 243-245: “However, we expect that our analyses show that current claims that growth of COVID-19 pandemics may be lower in developing tropical countries should be taken very carefully” Nevertheless there is less connection in developing tropical countries than in countries from the northern hemisphere? This low connectivity may also explain the lower development of the epidemics in some tropical countries.
The same remark can be made about “invested less in health”.

Figure 2: The readability of Fig 2 is low. I suggest to put Fig. 2A on a new Fig. 2 and the rest of Fig 2 and a new Fig. 3.

Experimental design

See below.

Validity of the findings

See below.

Additional comments

In this well written work the authors have used a classical macroecological approach to analyze the Covid-19 epidemics around the world. They have analyzed the growth rate at the beginning of the epidemic in numerous countries in relation with socioeconomic, climatic and transport variables. The key point of this study is the use of data from air traffic and the construction of the global air transportation network.

This study presents linear model analysis showing that the growth rate of the Covid-19 epidemic is mainly related to the airline connectivity between countries. With this kind of correlative analysis is important to propose potential explanations to the observed correlations. However the authors proposed no clear explanations to their main results, the effect of airline connectivity. This is one of the weaknesses of this work. Is this correlation observed, linked to the increase of the importation of severe cases? More discussion is needed on the role of air traffic. For instance, at the early beginning of the pandemic, the group of Colizza (Pullano et al., 2020. Novel coronavirus (2019-nCoV) early-stage importation risk to Europe, January 2020. Eurosurveillance, 25(4), 2000057.) has analyzed the risk of importation of the SARS-COV-19 in the European countries due to air traffic. The results showed that UK, France and Germany were the 3 countries with the higher risk of importation of this virus. Nevertheless, in real life, the epidemic started first and was strongest in Spain and Italy and started later in UK, which shows that the flow of passengers between airports is not always very reliable for making epidemiological predictions (similar results have been observed for the Ebola crisis).

One of my other concerns would be to better clarify the new contributions of this study compared to those already published on this topic (Pullano et al., 2020; Kraemer et al., 2020; Chinazzi et al., 2020; …).

Another of my concerns is about the fact that most of the Discussion is on the results of the influence of the warmer and humid climate on the propagation on tropical countries. It seems that the design of the statistical analysis is done, in the current work, for all the countries with climatic explicative variables (Mean Temperature and Mean Precipitation). With this analysis they cab only conclude that these 2 climatic variables have mild or no effect on the growth rate at the beginning of the Covid-19 epidemic. If the authors wanted to analyze the effect of climatic variables on the Covid-19 epidemic in tropical countries they would have to focus on the sub-set of tropical countries with other similar confounding factors. This concern is also true for the Abstract and the Conclusion.

To conclude, in this special time of crisis with an impressive flow of papers on Covid-19, it would be important that the authors associate their correlative with results some potential explanations that could convince epidemiologists. This is an important aspect that needed to be improved upon before publication in a journal with a large audience.

---

## Round 0.2 · accepted · Accept

I believe authors have succesfully replied to each and every critique made by the reviewers and myself. I'm glad one of the comments by R2 made you re-analyse the data and find errors with commas and points.
Figures are much improved now and I think the current version of the manuscript is ready to be published.